# Four New Species of Small-Spored *Alternaria* Isolated from *Solanum tuberosum* and *S. lycopersicum* in China

**DOI:** 10.3390/jof9090880

**Published:** 2023-08-27

**Authors:** Yanan Gou, Sein Lai Lai Aung, Zhuanjun Guo, Zhi Li, Shulin Shen, Jianxin Deng

**Affiliations:** 1Department of Plant Protection, College of Agriculture, Yangtze University, Jingzhou 434025, China; gynan024@hotmail.com (Y.G.); seinlailaiaung.ppd@gmail.com (S.L.L.A.); guozj13@hotmail.com (Z.G.); lz549440@hotmail.com (Z.L.); Shen33351@outlook.com (S.S.); 2MARA Key Laboratory of Sustainable Crop Production in the Middle Reaches of the Yangtze River (Co-Construction by Ministry and Province), Yangtze University, Jingzhou 434025, China

**Keywords:** small-spored *Alternaria*, *S. tuberosum*, *S. lycopersicum*, taxonomy, pathogenicity

## Abstract

Small-spored *Alternaria* species have been frequently isolated from diseased leaves of *Solanum* plants. To clarify the diversity of small-spored *Alternaria* species, a total of 118 strains were obtained from leaf samples of *S. tuberosum* and *S. lycopersicum* in six provinces of China during 2022–2023. Based on morphological characterization and multi-locus phylogenetic analysis of the internal transcribed spacer of the rDNA region (ITS), glyceraldehyde-3-phosphate dehydrogenase (*GAPDH*), translation elongation factor 1 alpha (*TEF1*), RNA polymerase second largest subunit (*RPB2*), *Alternaria* major allergen gene (*Alt a 1*), endopolygalacturonase gene (*EndoPG*) and an anonymous gene region (OPA10-2), seven species were determined, including four novel species and three known species (*A. alternata*, *A. gossypina* and *A. arborescens*). The novel species were described and illustrated as *A. longxiensis* sp. nov., *A. lijiangensis* sp. nov., *A. lycopersici* sp. nov. and *A. solanicola* sp. nov.. In addition, the pathogenicity of the seven species was evaluated on potato leaves. The species exhibited various aggressiveness, which could help in disease management.

## 1. Introduction

The plants of the Solanaceae family are widely cultivated worldwide and have significant economic, medicinal and ornamental values. There are approximately 115 species belonging to 25 genera in China [1,2,3]. For example, potatoes, peppers, tomatoes, eggplants, tobacco, etc. are important cash crops. *Solanum* is the largest genus, including more than 2000 plants including potato and tomato. (http://www.iplant.cn/) (accessed on 5 July 2023) As the fourth most important crop after rice, wheat and maize, potato (*S. tuberosum*) has become a staple food in many European countries because of its high nutritional value, rich in starch as well as protein, multivitamins and trace elements [4,5,6]. China’s potato acreage and its total production have consistently held a leading position in the world [7,8]. Tomato (*S. lycopersicum*) is a popular fruit and vegetable containing abundant vitamin C, cultivated all over the world [9]. Tomato fruit can be eaten raw, cooked, in processed ketchup, juice and so on, whose plant can also be used as potted ornamentals [10]. Moreover, it has great medicinal properties, of which lycopene, a powerful antioxidant may protect against cardiovascular disease and some cancers [11].

The two *Solanum* crops are susceptible to fungal pathogen infection during their cultivation, which severely threatens their yield and quality [12,13]. Especially for *Alternaria* spp., they are widely distributed with a great impact on economic returns of crops. When encountering favorable conditions, it can cause serious yield reduction or even no harvest [14,15]. Alternaria foliar disease caused by large-spored *Alternaria* species has been frequently reported from section (sect.) *Porri* comprising *A*. *solani*, *A*. *blumeae*, *A*. *linariae*, *A*. *grandis* and *A*. *protenta* infecting both potatoes and tomatoes [14,16]. Another type of *Alternaria*, small-spored *Alternaria* has been detected in a large number of the sampled leaves [17]. On potato, *A. alternate* (syn. *A. tenuissima*, *A. dumosa* and *A. interrupta*), *A. arborescens* and *A. longipes* in sect. *Alternaria* have been reported [5,17,18]. Most of the small-spored *Alternaria* species on tomato are *A. alternata*, *A. arborescens* and *A. tomato* [11,19]. In addition, *A. infectoria* in sect. *Infectoriae* is also often encountered from both plants [10,18].

Accurate identification methods are essential for the taxonomy of small-spored *Alternaria*. The morphological taxonomy is primarily based on the morphological traits of conidia and sporulation patterns [20,21]. Since the 20th century, molecular approaches have been utilized to identify *Alternaria* species [22,23,24,25]. At present, species identification based on multi-locus phylogenetic analysis has been applied as a powerful classification tool to assist morphological taxonomy, which has resulted in a combination of both approaches to determine these species [20,26,27,28].

During a survey of small-spored *Alternaria* species in China, the conidia continue to be discovered on diseased leaf samples of potato and tomato. The pre-test of the species diversity on both *Solanum* plants revealed four novel species and three known species (*A. alternata*, *A. gossypina* and *A. arborescens*). The aim of this study was to describe those four new taxa according to morphological traits and multi-locus phylogenetic analysis. In addition, the pathogenicity comprising the known species was evaluated on potato here.

## 2. Materials and Methods

### 2.1. Sampling and Isolation

Symptomatic samples of potato and tomato resembling Alternaria leaf spot or blight were randomly collected from 6 provinces in 2022–2023 (Table 1). The samples were preserved in sterile plastic bags and taken to laboratory for further isolation. The leaf fragments from edge of the lesions were cut and placed on moist filter papers in Petri dishes and incubated at 25 °C in dark forsporulation. Single spore was picked using a sterile glass needle under a stereomicroscope and inoculated onto potato dextrose agar (PDA: Difco, Montreal, Canada). Strains were kept into test-tube slants stored at 4 °C in the Fungal Herbarium of Yangtze University (YZU), Jingzhou, Hubei, China.

### 2.2. Morphology and Culture Characteristics

Strains representing each species were selected and cultured on PDA at 25 °C in dark for 7 days to determine the cultural features. To confirm the conidial morphology (conidial size, shape, sporulation patterns, etc.), fresh fungal mycelia were transferred onto potato carrot agar (PCA) and V8 juice agar (V8A) media, and then incubated at 22 °C with 8 h light/16 h dark period [21]. After 7 days, conidia and sporulation patterns were examined and photographed with a Nikon Eclipse Ni-U microscope system (Nikon, Tokyo, Japan). Fifty randomly selected conidia were measured for each strain.

### 2.3. DNA Extraction and PCR Amplification

Genomic DNA was extracted from fresh mycelium scraped from colonies on PDA using the CTAB method described in Watanabe et al. [29]. There were seven gene regions including internal transcribed spacer of the rDNA region (ITS), glyceraldehyde-3-phosphate dehydrogenase (*GAPDH*), translation elongation factor 1 alpha (*TEF1*), RNA polymerase second largest subunit (*RPB2*), *Alternaria* major allergen gene (*Alt a 1*), endopolygalacturonase (*EndoPG*) gene and an anonymous gene region (OPA10-2) used for phylogenic analyses. Polymerase chain reaction (PCR) amplifications were performed with the primer pairs of ITS5/ITS4 [30], gpd1/gpd2 [31], EF1-728F/EF1-986R [32], RPB2-5F/RPB2-7cR [33], Alt-for/Alt-rev [23], PG3/PG2b [34] and OPA10-2L/OPA10-2R [34], respectively. The PCR reaction was performed in a 25 μL volume containing 21 μL 1.1× Taq PCR Star Mix (TSINGKE, Beijing, China), 2 μL template DNA, and 1 μL of each primer. The amplified program for PCR amplifications was referenced from Woudenberg et al. [20]. Then, successful amplified products were purified and sequenced by TSINGKE company (Beijing, China). The sequences were deposited in GenBank (https://www.ncbi.nlm.nih.gov/) (accessed on 15 May 2023) and the accession numbers are shown in Table 2.

### 2.4. Phylogenetic Analysis

The resulting sequences were checked by the BioEdit v. 7.2.3 [35] and primarily aligned using the program of PHYDIT v.3.2 [36]. Phylogenetic analysis of RPB2 gene for all strains was performed for pre-test. Then, each of ITS, *GAPDH*, *TEF1*, *RPB2*, *Alt a 1*, *EndoPG* and OPA10-2 gene sequences was analyzed by BLAST search in NCBI (https://www.ncbi.nlm.nih.gov/) (accessed on 15 May 2023). Their relevant sequences were downloaded from the GenBank database. All gene sequences were spliced and edited through manual processing in MEGA v.7.0 [37]. The multi-locus phylogenetic trees were constructed based on Bayesian inference (BI) and maximum likelihood (ML) analyses. 

The ML analyses were performed using RAxML v.7.2.8 [38] using the GTRCAT model and bootstrapping with 1000 replicates. MrModeltest v.2.3 [39] used the Akaike Information Criterion (AIC) to determine the best-fit model (GTR + I + G) of nucleotide substitution, which was used for the Bayesian analyses performed with MrBayes v. 3.1.2 [40]. The analyses of two simultaneous Markov Chain Monte Carlo (MCMC) chains were run from random trees for 10, 000, 000 generations and sampled every 100th generations. The first 25% of the samples were discarded. Finally, the resulting trees were edited in FigTree v.1.3.1. [41]. Branch support of the analysis (>60%/0.6 for ML bootstrap value-BS/posterior probability-PP) was indicated in the phylogram.

### 2.5. Pathogenicity Tests

Twelve representative strains were selected to determine the pathogenicity of those present identified species, among which the four new species did not induce symptoms on tomato according to pre-tests except potato. Hence, the local grown potato cultivars were transplanted in pots and grown in greenhouse (25 °C, 12 h light period) for two weeks used for the experiment. The *Alternaria* strains were cultured on PDA at 25 °C for 3–5 days, and a 6 mm diameter disc was obtained from the colony edges and inoculated on healthy living leaves sterilized with 70% ethanol in the same greenhouse. The disease development was observed daily. After 7 days, the developed symptoms were recorded and the disease lesion size (LS) was measured. Control experiments were carried out simultaneously using clean PDA discs. For each strain, two potato plants were used and three sites were inoculated for each plant. To maintain the accuracy of the results, potato leaves of the same growing period with uniform size were selected for each test. The treatment was conducted four times. To complete Koch’s postulates, the same inoculated *Alternaria* was successfully reisolated from their induced symptoms for morphology identification and *RPB2* gene sequence analysis. The LS values were the mean value of four replicates ± standard deviation. The least significant difference test (*p* < 0.05) was conducted using IBM SPSS Statistics 23 for analysis.

## 3. Results

### 3.1. Phylogenetic Analysis 

A total of 118 small-spored *Alternaria* strains were obtained according to morphological traits. The preliminary *RPB2* gene sequence analysis showed that all strains belonged to sect. *Alternaria*, of which 88.13% was *A. alternata* and the others appeared as six taxa with different culture characteristics. Two strains of *A. alternata* and strains of the six taxa were further determined using the other six gene loci. Phylogenetic analysis included 88 strains from sect. *Alternaria* (Table 2), which comprised both reference strains and the present strains. The analysis was based on a combined ITS, *GAPDH*, *TEF1*, *RPB2*, *Alt a 1*, OPA10-2 and *EndoPG* sequence dataset, which included 494, 501, 215, 596, 437, 627 and 442 characters after alignment, respectively. *Alternaria alternantherae* (CBS 124392) from sect. *Alternantherae* was chosen as the outgroup taxon. The BI and ML analyses exhibited similar topologies. The ML tree was used as the basal tree (Figure 1). The result was similar to the *RPB2* gene phylogram. Four selected strains were well merged into the clades of three known *Alternaria* species, *A. alternata*, *A. arborescens* and *A. gossypina*, supported with PP/BS values of 0.97/79, 0.99/99 and 0.95/98, respectively. The other eight strains fell into four well independent clades supported with 0.99–1.00 PP values and 99–100% BP values, close to *A. orobanches* and *A. ovoidea* [42,43] in a branch supported with 0.80/73 PP/BS values. The results indicate that the four clades represent four new species.

**Table 2 jof-09-00880-t002:** *Alternaria* strains used in this study and their GenBank accession numbers (The present strains in bold).

Species	Strain	Host/Substrate	Country	GenBank Accession Number						
				ITS	GAPDH	TEF1	RPB2	Alt a 1	EndoPG	OPA10-2
*A. alternantherae*	CBS 124392	*Solanum melongena*	China	KC584179	KC584096	KC584633	KC584374	KP123846	–	–
*A. alternata*	CBS 916.96 T	*Arachis hypogaea*	India	AF347031	AY278808	KC584634	KC584375	AY563301	JQ811978	KP124632
	CBS 119399	*Minneola tangelo*	USA	KP124361	KP124063	KP125137	JQ646328	KP123910	KP124829	KP124672
	CBS 102604	*Minneola tangelo*	Israel	KP124334	AY562410	KP125110	KP124802	AY563305	KP124035	KP124643
	CBS 102602	*Minneola tangelo*	Turkey	KP124332	KP124187	KP125108	KP124800	KP123881	AY295023	KP124641
	CBS 102599	*Minneola tangelo*	Turkey	KP124330	KP124185	KP125106	KP124798	KP123879	KP124032	KP124639
	CBS 102595	*Citrus jambhiri*	USA	FJ266476	AY562411	KC584666	KC584408	AY563306	KP124029	KP124636
	CBS 127672	*Astragalus bisulcatus*	USA	KP124382	KP124234	KP125160	KP124852	KP123930	KP124086	KP124695
	CBS 103.33	Soil	Egypt	KP124302	KP124159	KP125077	KP124770	KP123852	KP123999	KP124607
	CBS 102.47	*Citrus sinensis*	USA	KP124303	KP124160	KP125079	KP124772	KP123854	KP124001	KP124609
	CBS 117.44	*Godetia* sp.	Denmark	KP124303	KP124160	KP125079	KP124772	KP123854	KP124001	KP124609
	CBS 102596	*Citrus jambhiri*	USA	KP124328	KP124183	KP125104	KP124796	KP123877	KP124030	KP124637
	CBS 918.96	*Dianthus chinensis*	UK	AF347032	AY278809	KC584693	KC584435	AY563302	KP124026	KP124633
	CBS 121455	*Broussonetia papyrifera*	China	KP124368	KP124220	KP125146	KP124838	KP123916	KP124072	KP124681
	CBS 121547	*Pyrus bretschneideri*	China	KP124372	KP124224	KP125150	KP124842	KP123920	KP124076	KP124685
	CBS 127671	*Stanleya pinnata*	USA	KP124381	KP124233	KP125159	KP124851	KP123929	KP124085	KP124694
	CBS 106.34	*Linum usitatissimum*	Unknown	Y17071	JQ646308	KP125078	KP124771	KP123853	KP124000	KP124608
	CBS 121336	*Allium* sp.	USA	KJ862254	KJ862255	KP125141	KP124833	KJ862259	KP124067	KP124676
	CBS 119543	*Citrus paradisi*	USA	KP124363	KP124215	KP125139	KP124831	KP123911	KP124065	KP124674
	CBS 195.86	*Euphorbia esula*	Canada	KP124317	KP124173	KP125093	KP124785	JQ646398	KP124017	KP124624
	**YZU 221086**	** *Solanum tuberosum* **	**China**	**OR437981**	**OR462188**	**OR455803**	**OR462192**	**OR455821**	**OR455823**	**OR455847**
	**YZU 221102**	** *Solanum tuberosum* **	**China**	**OR437980**	**OR462187**	**OR455802**	**OR462191**	**OR455820**	**OR455822**	**OR455846**
	CBS 479.90	*Citrus unshiu*	Japan	KP124319	KP124174	KP125095	KP124787	KP123870	KP124019	KP124626
*A. arborescens*	CBS 101.13	Peat soil	Switzerland	KP124392	KP124244	KP125170	KP124862	KP123940	KP124096	KP124705
	CBS 119545 T	*Senecio skirrhodon*	New Zealand	KP124409	KP124260	KP125187	KP124879	KP123956	KP124113	KP124723
	CBS 119544 T	*Avena sativa*	New Zealand	KP124408	JQ646321	KP125186	KP124878	KP123955	KP124112	KP124722
	CBS 112749	*Malus domestica*	South Africa	KP124402	KP124254	KP125180	KP124872	KP123949	KP124106	KP124716
	CBS 105.24	*Solanum tuberosum*	Unknown	KP124393	KP124245	KP125171	KP124863	KP123941	KP124097	KP124706
	CBS 126.60	Wood	UK	KP124397	KP124249	KP125175	KP124867	JQ646390	KP124101	KP124710
	CBS 109730	*Solanum lycopersicum*	USA	KP124399	KP124251	KP125177	KP124869	KP123946	KP124103	KP124713
	CBS 105.49	Contaminant blood culture	Italy	KP124396	KP124248	KP125174	KP124866	KP123944	KP124100	KP124709
	**YZU 221451**	** *Solanum tuberosum* **	**China**	**OR435100**	**OR270928**	**OR270930**	**OR462190**	**OR455819**	**OR270925**	**OR270924**
*A. baoshanensis*	MFLUCC 21-0124 T	*Curcubita moschata*	China	MZ622003	OK236706	OK236613	OK236659	OK236760	–	–
	MFLU 21-0296	*Curcubita moschata*	China	MZ622004	OK236707	OK236612	OK236660	OK236759	–	–
*A. betae-kenyensis*	CBS 118810	*Beta vulgaris* var. *cicla*	Kenya	KP124419	KP124270	KP125197	KP124888	KP123966	KP124123	KP124733
*A. breviconidiophora*	MFLUCC 21-0786 T	*Digitalis* sp.	Italy	MZ621997	OK236698	OK236604	OK236651	OK236751	–	–
	MFLU 21-0317	*Digitalis* sp.	Italy	MZ621998	OK236699	OK236605	OK236652	OK236752	–	–
*A. burnsii*	CBS 118816	*Rhizophora mucronata*	India	KP124423	KP124273	KP125201	KP124892	KP123970	KP124127	KP124737
	CBS 118817	*Tinospora cordifolia*	India	KP124424	KP124274	KP125202	KP124893	KP123971	KP124128	KP124738
*A. eichhorniae*	CBS 489.92 T	*Eichhornia crassipes*	India	KC146356	KP124276	KP125204	KP124895	KP123973	KP124130	KP124740
*A. ellipsoidialis*	MFLUCC 21-0132 T	*Brassica* sp.	Italy	MZ621989	OK236690	OK236596	OK236643	OK236743	–	–
	MFLU 21-0307A	*Brassica* sp.	Italy	MZ621990	OK236691	OK236597	OK236644	OK236744	–	–
*A. eupatoriicola*	MFLUCC 21-0122 T	*Eupatorium cannabinum*	Italy	MZ621982	OK236683	OK236589	OK236636	OK236736	–	–
	MFLU 21-0319	*Eupatorium cannabinum*	Italy	MZ621983	OK236684	OK236589	OK236637	OK236737	–	–
*A. falcata*	MFLUCC 21-0123 T	*Atriplex* sp.	Italy	MZ621992	OK236693	OK236599	OK236649	OK236746	–	–
	MFLU 21-0306	*Atriplex* sp.	Italy	MZ621993	OK236694	OK236600	OK236650	OK236747	–	–
*A. gaisen*	CBS 632.93 R	*Pyrus pyrifolia*	Japan	KC584197	KC584116	KC584658	KC584399	KP123974	AY295033	KP124742
	CBS 118488 R	*Pyrus pyrifolia*	Japan	KP124427	KP124278	KP125206	KP124897	KP123975	KP124132	KP124743
*A. gossypina*	CBS 104.32 T	*Gossypium* sp.	Zimbabwe	KP124430	KP124135	KP125209	JQ646312	JQ646395	KP124900	KP124746
	CBS 102597	*Minneola tangelo*	USA	KP124432	KP124281	KP125211	KP124902	KP123978	KP124137	KP124748
	CBS 102601	*Minneola tangelo*	Colombia	KP124433	KP124282	KP125212	KP124903	KP123979	KP124138	KP124749
	**YZU 221455**	** *Solanum tuberosum* **	**China**	**OR435098**	**OR270927**	**OR270929**	**OR462189**	**OR455818**	**OR270926**	**OR270923**
*A. iridiaustralis*	CBS 118486 T	*Iris* sp.	Australia	KP124435	KP124284	KP125214	KP124905	KP123981	KP124140	KP124751
	CBS 118487	*Iris* sp.	Australia	KP124436	KP124285	KP125215	KP124906	KP123982	KP124141	KP124752
*A. jacinthicola*	CBS 133751 T	*Eichhornia crassipes*	Mali	KP124438	KP124287	KP125217	KP124908	KP123984	KP124143	KP124754
	CBS 878.95	*Arachis hypogaea*	Mauritius	KP124437	KP124286	KP125216	KP124907	KP123983	KP124142	KP124753
** *A. lijiangensis* **	**YZU 221458**	** *Solanum tuberosum* **	**China**	**OQ679970**	**OQ686785**	**OQ686783**	**OQ686789**	**OQ686781**	**OQ686779**	**OQ686787**
	**YZU 221459**	** *Solanum tuberosum* **	**China**	**OQ679971**	**OQ686786**	**OQ686784**	**OQ686790**	**OQ686782**	**OQ686780**	**OQ686788**
*A. longipes*	CBS 540.94	*Nicotiana tabacum*	USA	AY278835	KP124147	KC584667	AY278811	AY563304	KC584409	KP124758
	CBS 121333 R	*Nicotiana tabacum*	USA	KP124444	KP124150	KP125223	KP124293	KP123990	KP124914	KP124761
** *A. longxiensis* **	**YZU 221221**	** *Solanum tuberosum* **	**China**	**OQ534546**	**OQ512732**	**OQ512726**	**OQ543009**	**OQ473629**	**OQ512720**	**OQ543003**
	**YZU 221222**	** *Solanum tuberosum* **	**China**	**OQ534547**	**OQ512731**	**OQ512725**	**OQ543008**	**OQ473628**	**OQ512719**	**OQ543002**
** *A. lycopersici* **	**YZU 221185**	** *Solanum lycopersicum* **	**China**	**OQ519795**	**OQ512736**	**OQ512730**	**OQ543013**	**OQ473633**	**OQ512724**	**OQ543007**
	**YZU 221186**	** *Solanum lycopersicum* **	**China**	**OQ519794**	**OQ512735**	**OQ512729**	**OQ543012**	**OQ473632**	**OQ512723**	**OQ543006**
*A. macroconidia*	MFLUCC 21-0134 T	*Spartium junceum*	Italy	MZ622001	OK236704	OK236610	OK236657	OK236757	–	–
	MFLU 21-0301	*Spartium junceum*	Italy	MZ622002	OK236705	OK236611	OK236658	OK236758	–	–
*A. minimispora*	MFLUCC 21-0127 T	*Citrullus lanatus*	Thailand	MZ621980	OK236681	OK236587	OK236634	OK236734	–	–
	MFLU 21-0318	*Citrullus lanatus*	Thailand	MZ621981	OK236682	OK236588	OK236635	OK236735	–	–
*A. muriformispora*	MFLUCC 21-0784 T	*Plantago* sp.	Italy	MZ621976	OK236677	OK236583	OK236630	OK236730	–	–
	MFLU 21-0309	*Plantago* sp.	Italy	MZ621977	OK236678	OK236584	OK236631	OK236731	–	–
*A. obpyriconidia*	MFLUCC 21-0121 T	*Vicia faba*	Italy	MZ621978	OK236680	OK236585	OK236633	OK236732	–	–
	MFLU 21-0300	*Vicia faba*	Italy	MZ621979	OK236679	OK236586	OK236632	OK236733	–	–
*A. orobanches*	MFLUCC 21-0137 T	*Orobanche* sp.	Italy	MZ622007	OK236710	–	–	OK236763	–	–
	MFLU 21-0303	*Orobanche* sp.	Italy	MZ622008	OK236711	–	–	OK236764	–	–
*A. ovoidea*	MFLUCC 21-0782 T	*Dactylis glomerata*	Italy	MZ622005	OK236708	OK236614	OK236661	OK236761	–	–
	MFLU 21-0298	*Dactylis glomerata*	Italy	MZ622006	OK236709	OK236615	OK236662	OK236762	–	–
*A. phragmiticola*	MFLUCC 21-0125 T	*Phragmites* sp.	Italy	MZ621994	OK236696	OK236602	OK236649	OK236749	–	–
	MFLU 21-0136	*Phragmites* sp.	Italy	MZ621995	OK236697	OK236603	OK236650	OK236750	–	–
*A. rostroconidia*	MFLUCC 21-0136 T	*Arabis* sp.	Italy	MZ621969	OK236670	OK236576	OK236623	OK236723	–	–
	MFLU 21-0299	*Arabis* sp.	Italy	MZ621970	OK236671	OK236577	OK236624	OK236724	–	–
*A. salicicola*	MFLUCC 22-0072 T	*Salix alba*	Russia	MZ621999	OK236700	OK236606	OK236653	OK236753	–	–
	MFLU 21-0320	*Salix alba*	Russia	MZ622000	OK236701	OK236607	OK236654	OK236754	–	–
** *A. solanicola* **	**YZU 221189**	** *Solanum lycopersicum* **	**China**	**OQ534548**	**OQ512734**	**OQ512728**	**OQ543011**	**OQ473631**	**OQ512722**	**OQ543005**
	**YZU 221190**	** *Solanum lycopersicum* **	**China**	**OQ519793**	**OQ512733**	**OQ512727**	**OQ543010**	**OQ473630**	**OQ512721**	**OQ543004**
*A. tomato*	CBS 114.35	*Solanum lycopersicum*	Unknown	KP124446	KP124295	KP125225	KP124916	KP123992	KP124152	KP124763
	CBS 103.30	*Solanum lycopersicum*	Unknown	KP124445	KP124294	KP125224	KP124915	KP123991	KP124151	KP124762
*A. torilis*	MFLUCC 21-0133	*Torilis arvensis*	Italy	MZ621986	OK236687	OK236593	OK236640	OK236740	–	–
	MFLU 21-0299	*Torilis arvensis*	Italy	MZ621987	OK236689	OK236595	OK236642	OK236742	–	–

### 3.2. Morphology

The morphological characteristics were measured and compared (Table 3), which was consistent with the result of multi-loci analysis. The four new species (Figure 2, Figure 3, Figure 4 and Figure 5) are illustrated and described.

### 3.3. Taxonomy


***Alternaria lycopersici* Y.N. GOU & J.X. Deng, sp. nov. YZU 221186**
** (Figure 2)**


MycoBank No.: 848427

**Etymology:** Named after the host species name, *Solanum lycopersicum*. 

**Typification:** China, Hubei Province, Jingzhou City, from leaf spot of *Solanum lycopersicum*, 1 July 2022, J.X. Deng, (YZU-H-2022047, holotype), ex-type culture YZU 221186. 

**Description:** Colonies on PDA circular, light cottony and buff in the center, villiform with white at the edge, 44–45.3 mm in diam., at 25 °C for 7 days. On PCA, conidiophores arising from substrate or lateral of aerial hyphae, cylindrical, straight or curved, septate, pale brown; 28–58(–68.5) × 3–5 μm (av.: 42 × 4 μm); conidia 2–4 units per chain, yellow-brown to light olive green with almost smooth-walled, straight, clavate, long ellipsoid or ovoid, 18–41 × 9.5–13 μm (av.: 31× 11 μm), 1–7 transverse septa, 0–2(−3) longitudinal septa, with an apical 1-cell secondary conidiophore (beak) around 3.5–8 μm in length. On V8A, conidiophores straight or curved, smooth–walled, septate, 26.5–59(−67) × 3.5–5 μm (av.: 41 × 4 μm), conidia 2–4 units in a chain, medium yellow-brown to light olive green, almost smooth ellipsoid or ovoid, 18.5–42.5 × 9–14 μm (av.: 30 × 9.5 μm), 2–7 transverse septa, 0–2(−3) longitudinal septa, with a small apical beak (3.5–8 μm long). 

**Notes:** Based on the combined dataset of ITS, *GAPDH*, *TEF1*, *RPB2*, *Alt a 1*, *EndoPG* and OPA10-2 gene fragments, the results reveal that the strains fall in an individual well-supported clade representing a new species, which is closer to *A. longxiensis* sp. nov. and *A. solanicola* sp. nov., near to *A. orobanches*. After a nucleotide pairwise comparison, the present species can be readily differentiated from the other two related novel species based on *Alt a 1, RPB2* and OPA10-2 gene regions, which has 21 bp differences in the *Alt a 1* region, 17 bp in *RPB2* and 19 bp in OPA10-2 when compared with *A. longxiensis*.

For *A. solanicola* sp. nov., there are 18 bp nucleotide differences in *Alt a 1* region, 16 bp in *RPB2* and 20 bp in the OPA10-2 region. Morphologically, its conidia are smallest, clavate and clearly smooth-walled with shorter apical beaks compared to the other two new species. (Figure 2, Table 3).

**Figure 2 jof-09-00880-f002:**
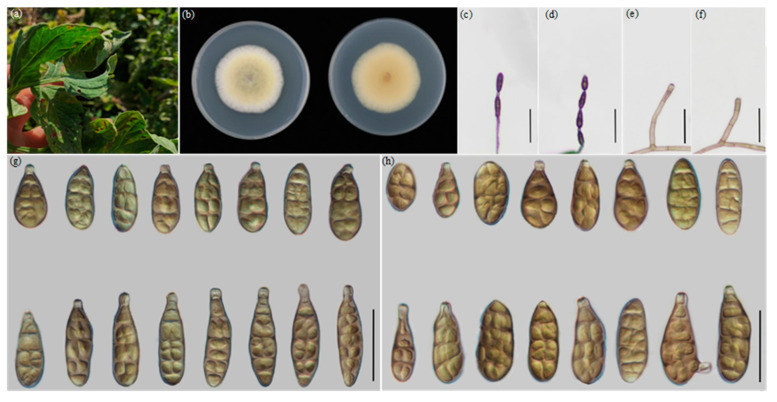
Morphology of *A. lycopersici* sp. nov.: (**a**) Diseased samples in field; (**b**) Colony phenotypes (on PDA for 7 days at 25 °C); (**c**,**d**) Sporulation patterns; (**e**,**f**) Conidiophores; (**g**) Conidia (on PCA at 22 °C; (**h**) Conidia (on V8A at 22 °C). Bars: (**c**,**d**) = 50 μm; (**e**–**h**) = 25 μm.


***Alternaria solanicola* Y.N. GOU & J.X. Deng, sp. nov. YZU 221190 (Figure 3)**


MycoBank No.: 848426

**Etymology:** Named after the host genus names, *Solanum lycopersicum*.

**Typification:** China, Hubei Province, Jingzhou City, from leaf spot of *Solanum lycopersicum*, 1 July 2022, J.X. Deng, (YZU-H-2022048, holotype), ex-type culture YZU 221190. 

**Description:** Colonies on PDA circular, pistac, with white in the center, ivory in reverse; 55.1– 56.6 mm in diam., at 25 °C for 7 days. On PCA, conidiophores arising from substrate, straight or curved, smooth-walled, septate, brown; 20.5–52 (–67) × 3–5 μm (av.: 38.5 × 3.5 μm); conidia 2–4 units per chain, pale to dark brown, almost smooth-walled, short to long ovoid or ellipsoid, 22–44 × 9–16.5 μm (av.: 32× 12 μm), 1–6 transverse septa, 0–2 longitudinal septa, without or with beak around 6–26.5 μm in length. On V8A, conidiophores straight to slightly curved, smooth-walled, septate, brown, 22–51.5(–65) × 3–5 μm (av.: 37 × 4 μm); conidia 2–4 units in a chain, yellow to dark brown, almost smooth-walled, short to long ovoid or ellipsoid, 22–43.5 × 9–16 μm (av.: 32.5 × 11.5 μm), 1–6 transverse septa, 0–2(−3) longitudinal septa, beakless or with an apical beak (7–26 μm long). 

**Notes:** The species is phylogenetically recognized as a distinct species that forms a subclade with *A. longxiensis* sp. nov. in a branch containing *A. lycopersici* sp. nov. and *A. orobanches*. On the bases of the *RPB2*, OPA10-2 and *Alt a 1* gene sequences, this species, respectively, comprises 3 bp, 20 bp and 18 bp nucleotide differences from *A. longxiensis*. According to morphology, it can be differentiated from *A. longxiensis* by producing conidia with longer beaks and fewer septa and by comprising fewer spore units in a chain with 1–2 branches. (Figure 3, Table 3).

**Figure 3 jof-09-00880-f003:**
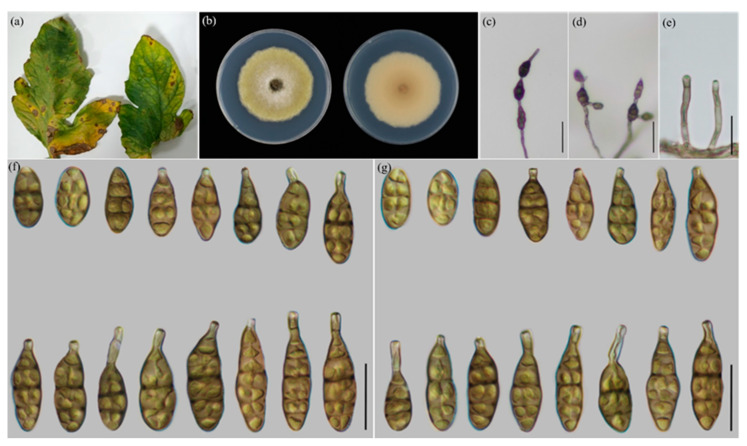
Morphology of *A. solanicola* sp. nov.: (**a**) Diseased samples from field; (**b**) Colony phenotypes (on PDA for 7 days at 25 °C); (**c**,**d**) Sporulation patterns; (**e**) Conidia (on PCA at 22 °C; (**f**) Conidia (on V8A at 22 °C). (**g**) Conidia (on V8A at 22 °C) Bars: (**c**,**d**) = 50 μm; (**e**–**g**) = 25 μm.


***Alternaria longxiensis* Y.N. GOU & J.X. Deng, sp. nov. YZU 221222 (Figure 4)**


MycoBank No.: 848428

**Etymology:** Named after the collecting locality, Longxi (Gansu, China). 

**Typification:** China, Gansu Province, Dingxi City, from leaf spot of *Solanum tuberosum*, 1 July 2022, J.X. Deng, (YZU-H-2022055, holotype), ex-type culture YZU 221222. 

**Description:** Colonies on PDA circular, light cottony and white to off-white in the center, villiform with white at the edge, reverse dark brown in the center, white at edges, 61.5–62.5 mm in diam., at 25 °C for 7 days. On PCA, conidiophores arising from substrate, simple, straight or flexuous, septate, light to dark brown; 25.5–67(–89.5) × 3–5.5 μm (av.: 46.5 × 4 μm); conidia 2–6 units per chain, medium yellow-brown to brown, short to long ellipsoid, or narrow-ovoid to ovoid, 22.5–51 × 8.5–17 μm (av.: 34 × 11 μm), 1–7 transverse septa, 1–2(−3) longitudinal septa, with an apical beak around 6–18.5 μm in length. On V8A, conidiophores straight or flexuous, septate, light to dark brown, 23–65.5(−87) × 3.5–5 μm (av.: 46 × 3.5 μm), conidia 2–6 units in a chain, medium yellow-brown to brown, short to long ellipsoid, or narrow-ovoid to ovoid, 22–51(–57) × 10–17.5 μm (av.: 35 × 14 μm), 1–7 transverse septa, 1–2(−3) longitudinal septa, with beak of 6–26.5 μm long.

**Notes:** For this species, the phylogenetic analysis shows that it is quite close to *A. solanicola* sp. nov. by grouping together in a well-supported subclade (PP/BS = 0.99/96). Morphologically, it produces larger conidia with shorter beaks and a longer conidial chain, when compared with *A. solanicola* sp. nov.. (Figure 4, Table 3).

**Figure 4 jof-09-00880-f004:**
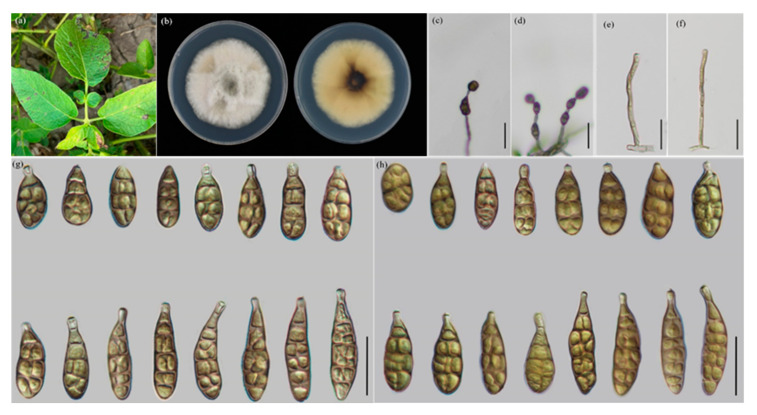
Morphology of *A. longxiensis* sp. nov.: (**a**) Diseased leaves in field; (**b**) Colony phenotypes (on PDA for 7 days at 25 °C); (**c**,**d**) Sporulation patterns; (**e**,**f**) Conidiophores (**g**) Conidia (on PCA at 22 °C; (**h**) Conidia (on V8A at 22 °C). Bars: (**c**,**d**) = 50 μm; (**e**–**h**) = 25 μm.


***Alternaria lijiangensis* YZU 221459 Y.N. GOU & J.X. Deng, sp. nov. (Figure 5)**


MycoBank No.: 848429

**Etymology:** Named after the collecting locality, Lijiang City. 

**Typification:** China, Yunnan Province, Lijiang City, from leaf spot of *Solanum tuberosum*, 7 Octorber 2022, J.X. Deng, (YZU-H-2022096, holotype), ex-type culture YZU 221459.

**Description:** Colonies on PDA circular, flocculent with white in the center, light yellow at margins, reverse dark brown in the center, 48.9–50.0 mm in diam., at 25 °C for 7 days. On PCA, conidiophores arising from substrate or lateral of aerial hyphae, straight or curved, septate, pale brown; 21.5–69 × 3–5 μm (av.: 43.5 × 4 μm); conidia 3–7 units per chain, yellow brown to dark brown, ovate, elliptic or ovoid to obpyriform, 25.5–45 × 10–15.5 μm (av.: 33 × 12.5 μm), 1–5 transverse septa, 0–2 longitudinal septa, with an apical extension (beak) around 5–40 μm. On V8A, conidiophores straight or curved, septate, 22–67 × 3.5–5 μm (av.: 44 × 3.5 μm), conidia 3–7 units in a chain, brown to dark brown, ovate, elliptic or ovoid to obpyriform, 22.5–44.5 × 11–15 μm (av.: 31.5 × 13 μm), 1–5 transverse septa, 0–2 longitudinal septa, with obtuse beak of 5–40 μm long. 

**Notes:** Phylogenetic analysis shows that the species falls into an independent lineage sister to the *A. orobanches*, *A. ovoidea* and three new species. Morphologically, it can be easily distinguished from the relevant species by producing conidia with the longest beak (up to 40 μm). (Figure 5, Table 3).

**Figure 5 jof-09-00880-f005:**
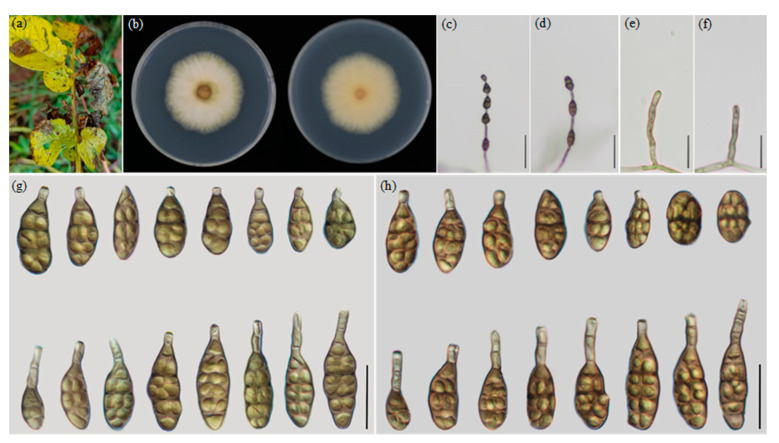
Morphology of *A. lijiangensis* sp. nov.: (**a**) Diseased leaves in field; (**b**) Colony phenotypes (on PDA for 7 days at 25 °C); (**c**,**d**) Sporulation patterns; (**e**,**f**) Conidiophores (**g**) Conidia (on PCA at 22 °C; (**h**) Conidia (on V8A at 22 °C). Bars: (**c**,**d**) = 50 μm; (**e**–**h**) = 25 μm.

### 3.4. Pathogenicity Assays

Pathogenicity tests indicated that the seven small-spored *Alternaria* species (*A. alternata*, *A. arborescens*, *A. gossypina*, *A. lycopersici*, *A. solanicola*, *A. longxiensis* and *A. lijiangensis*) of sect. *Alternaria* were all pathogenic to *S. tuberosum* and showed varying degrees of pathogenicity. No symptoms were observed in the controls. (Figure 6, Table 4) After three days, black spots began to appear on the leaves and gradually expand, accompanied by yellow halos. Strains of *A. lijiangensis* exhibited the most severe symptom with LS up to 30.5 mm (av. = 27.5) resulting in a whole dark brown leaf wilting. The disease severity was followed by *A. gossypina* (LS up to 24 mm) and *A. arborescens* with LS around 18 to 20 mm, and then *A. longxiensis* sp. nov., *A. lycopersici* sp. nov., and *A. solanicola* sp. nov.. The weakest pathogenicity on potato was *A. alternata* among all the tested species.

## 4. Discussion

Simmons dedicates to the morphological taxonomy of *Alternaria* using a life time. A total of 276 species are comprehensively illustrated based on sporulation patterns and conidial morphology, comprising 128 small-spored *Alternaria* species [21]. With the continuous research on the taxonomy of small-spored *Alternaria* aided with molecular approach, the sect. *Alternaria* is one of the largest sections containing 11 species and one species complex [20], of which *A. alternata* includes 35 morph-species described by Simmons [21]. There are 27 new species latterly defined as new members of sect. *Alternaria* [27,42,43,45]. In this study, four novel species (*A. lycopersici* sp. nov., *A. solanicola* sp. nov., *A. longxiensis* sp. nov. and *A. lijiangensis* sp. nov.) were as found in China and added in the section.

Phylogenetically, nine commonly used genetic regions (SSU, LSU, ITS, *GADPH*, *RPB2*, *TEF1*, *Alt a1*, *EndoPG*, and OPA10-2) have been used for the delamination of species within sect. *Alternaria* [46,47,48]. Among them, the *RPB2* gene is considered as a nuclear gene with the advantages of being single-copy and having a slow evolutionary rate. It is capable of effectively distinguishing species for both large-spored and small-spored type [49,50], and is also applicable for the identification of other pathogenic fungi [51]. In this present study, seven gene loci without SSU and LSU were performed and clearly separated the four new taxa. In addition, the *RPB2* gene was again confirmed for the ability of fungal classification, which resulted in similar consequences to distinguish the present seven *Alternaria* species (Figure 1). The present novel species fell into four independent clades together with *A. orobanches* and *A. ovoidea,* which were isolated from *Orobanche* sp. and *Dactylis glomerata* in Italy, respectively [42,43].

Morphologically, these four new species can be distinguished by their conidia and sporulation patterns. Among them, *A. lijiangensis* has conidia with the longest beaks among the four present novel species and its relevant species, *A. orobanches* and *A. ovoidea*. The other three new species had their own characteristics and could be differentiated with *A. orobanches* and *A. ovoidea* by sporulation pattern (more conidial units per chain). (Table 3) For example, conidia of *A. lycopersici* has a shorter apical cell secondary conidiophore (beak); conidia of *A. longxiensis* has a longer beak; *A. solanicola* has 1–2 branches near the main chain. The results indicated the correlation between morphology and molecular composition, and also stated the usefulness of morphological traits (conidial morphology and sporulation pattern) described by Simmons [20] for the taxonomy of small-spored *Altenraria*.

The large-spored *A. solani and A. linariae,* are considered to be the primary pathogen of foliar disease in the Solanaceae family [14,44,52], but small-spored *Alternaria* are also frequently isolated from symptomatic tissues worldwide. Nabahat et al. [17] found that species within sect. *Alternaria* were dominant populations in Solanaceae plants during their isolation process, accounting for more than 90% of isolated *Alternaria* strains, which is similar to the reports in the United States and Russia [18,53,54]. In addition to those four new small-spored *Alternaria*, three known species on potatoes, *A. alternata*, *A. gossypina* and *A. arborescens*, were also discovered (Table 1), of which *A. alternata* was a common species found on both hosts, detected in 91.21% on potato and 85.18% on tomato. *A. gossypina* has been commonly reported as the primary pathogen of leaf spot in cotton [55,56], but has not been found on potatoes worldwide, which is firstly found on potato (Appendix A, Table 3). *A. arborescens* is frequently reported worldwide on Solanaceae [57,58]. However, to the best of our knowledge, it is the first record on potato in China (Appendix A, Table 3).

For the pathogenicity tests (Figure 6, Table 4), *A. lijiangensis* sp. nov. showed the most severe symptoms with the LS up to 30.5 mm, which was found in Lijiang, Yunnan Province. Then, the *A. gossypina* was indicated to be the second most aggressive to potatoes (LS up to 24 mm), followed by *A. arborescens*, *A. longxiensis* sp. nov. and *A. lycopersici* sp. nov. and *A. solanicola* sp. nov.. Compared to the above species, *A. alternata* were weakest in pathogenicity. In addition, pathogenicity tests of the four new species were also conducted on detached and living tomato leaf with non-pathogenicity, but they exhibited a certain level of aggressiveness on potatoes. The two species from tomato, *A. lycopersici* sp. nov. and *A. solanicola* sp. nov. were not pathogenic to their host. It might be because both species were saprophytic or weakly pathogenic facing a resistant tomato variety when conducting the pre-test. According to Zheng et al. [5], *A. tenuissima, A. alternata* and *A. solani* are the main pathogens for the Alternaria foliar diseases on potato in China. According to previous study, it is better to be referred *A. tenuissima* as *A. alternata* [20]. The present finding showed that a diverse set of small-spored *Alternaria* species induced various symptoms on potato in China, which might be due to the changing climatic conditions [59,60]. The results indicated that small-spored *Alternaria* had a potential to threaten our potato production. Nabahat et al. [17] mentioned that *A. alternata* and *A. arborescens* could induce leaf blight and spot diseases on tomato and potato; when co-infected with moderately aggressive isolates of *A. linariae*, synergistic interactions might also occur. Thus, more samples will be collected nationally and more studies will be continued to help field disease management.

## Figures and Tables

**Figure 1 jof-09-00880-f001:**
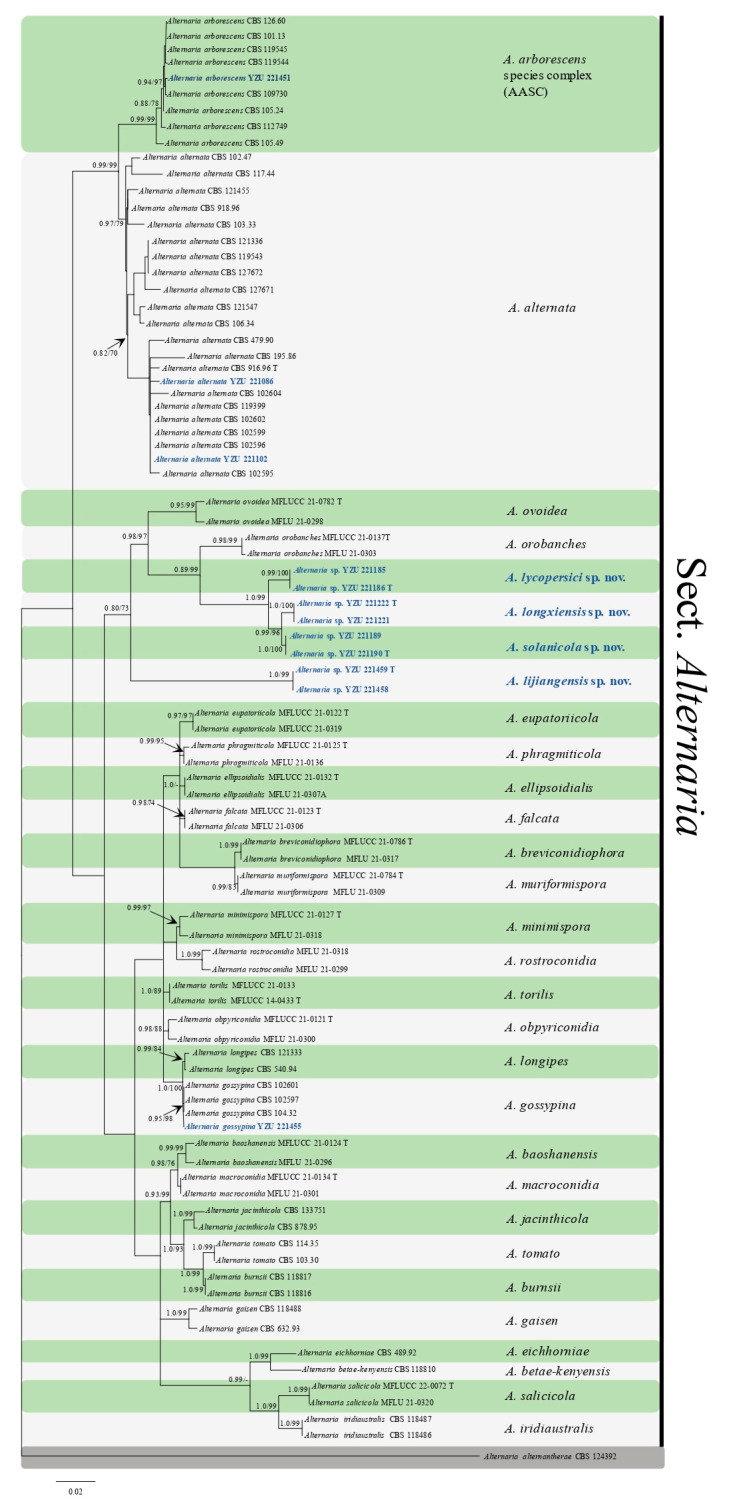
Phylogenetic tree based on the combined gene sequences of ITS, *GAPDH*, *TEF1*, *RPB2*, *Alt a 1*, *EndoPG* and OPA10-2 generated from *Alternaria* spp. on potatoes and tomatoes. The Bayesian posterior probabilities (PP > 0.60) and maximum likelihood bootstrap values (BS > 60%) are given at the nodes (PP/BS). Examined present strains are in bold.

**Figure 6 jof-09-00880-f006:**
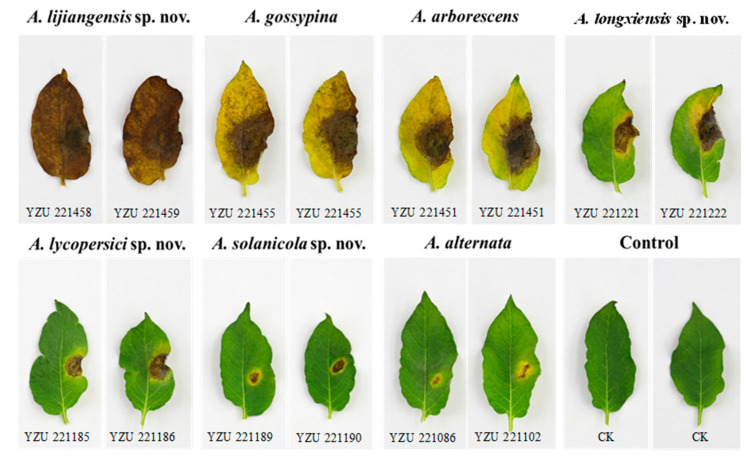
Pathogenicity of the seven present small-spored *Alternaria* species on *Solanum tuberosum*.

**Table 1 jof-09-00880-t001:** The numbers of *Alternaria* strains collected from potato and tomato in different provinces of China.

Host	Province	Samples	Total Number of Isolated Strains
*Solanum tuberosum*	Hubei	12	40
	Gansu	5	14
	Shaanxi	5	18
	Sichuan	2	4
	Yunnan	4	10
	Xinjiang	3	5
*Solanum lycopersicum*	Hubei	6	12
	Gansu	2	2
	Sichuan	2	2
	Xinjiang	4	11

**Table 3 jof-09-00880-t003:** Morphological comparisons of the present seven *Alternaria* species and their closely related species.

Species	Conidia				Sporulation Pattern		Medium /Host	Reference
	Shape	Body (μm)	Beak (μm)	Septa	Conidia Per chain	Branches		
*A. alternata*	Ovoid, ellipsoid, or subsphaeroid or with short beak	7–30(−40) × 5–12	3–5 (−30)	1–7	4–20	Multiple	PCA	[21]
** *A. alternata* **	**Ovoid or ellipsoid with cylindric apex**	**11–33 × 4–8**	**4–9 (−16)**	**17**	**3–9**	**0–3**	**PCA**	**This study**
*A. arborescens*	Short-ovoid or ellipsoid	12–30(–42) × 7–11	–	1–4	2–6	–	PCA	[21]
** *A. arborescens* **	**Short-ovoid or ellipsoid**	**15–32.5(–45) × 7–13**	**–**	**1–5**	**2–8**	**2–5**	**PCA**	**This study**
*A. gossypina*	Obclavate or ovoid	(14.5–)30.5–42.5(–48) × 11–14.5(−17)	7.5–33	5–9	–	–	Host	[44]
** *A. gossypina* **	**Obclavate or ovoid**	**(14–)28.5–54 ×10–16(−19)**	**6–35**	**3–8**	**5–11**	**0**	**PCA**	**This study**
*A. orobanches*	Obclavate to ovoid	20–50 × 10–20	–	3–6	1–2	–	PCA	[42]
*A. ovoidea*	Ovoid	48–65 × 15.5–30	–	1–3	1	–	PCA	[43]
** *A. lycopersici* **	**Straight, clavate, ellipsoid or ovoid**	**18–41 × 9.5–13**	**3.5–8**	**1–7**	**2–4**	**0**	**PCA**	**This study**
** *A. solanicola* **	**Short to long oviod or ellipsoid or with beak**	**22–44 × 9–16.5**	**6–26.5**	**1–6**	**2–4**	**0–2**	**PCA**	**This study**
** *A. longxiensis* **	**Short to long ellipsoid or narrow-ovoid**	**22.5–51 × 8.5–17**	**6–18.5**	**1–7**	**2–6**	**0**	**PCA**	**This study**
** *A. lijiangensis* **	**Ovate, elliptic or ovoid to obpyriform**	**25.5–45 × 10–15.5**	**5–40**	**1–5**	**3–7**	**0**	**PCA**	**This study**

**Table 4 jof-09-00880-t004:** Disease incidence and lesion size of seven present small-spored *Alternaria* species on *Solanum tuberosum*.

Species	Strain	Disease Incidence (%)	Lesion (mm)
***A. lijiangensis* sp. nov.**	YZU 221458	100 ± 0 a	27.5 ± 0.69 a
	YZU 221459	100 ± 0 a	27.04 ± 0.31 a
** *A. gossypina* **	YZU 221455	100 ± 0 a	21.35 ± 0.62 b
** *A. arborescens* **	YZU 221451	100 ± 0 a	18.14 ± 0.15 c
***A. longxiensis* sp. nov.**	YZU 221221	100 ± 0 a	15.65 ± 0.49 d
	YZU 221222	100 ± 0 a	16.35 ± 0.89 d
***A. lycopersici* sp. nov.**	YZU 221185	100 ± 0 a	11.27 ± 0.34 e
	YZU 221186	100 ± 0 a	11.5 ± 0.20 e
***A. solanicola* sp. nov.**	YZU 221189	100 ± 0 a	8.63 ± 0.39 f
	YZU 221190	100 ± 0 a	8.22 ± 0.88 f
** *A. alternata* **	YZU 221086	100 ± 0 a	6.75 ± 0.08 g
	YZU 221102	100 ± 0 a	6.08 ± 0.45 g

**Notes:** Disease incidence (DI) was evaluated by counting the percentage of diseased leaves. Lesion size (LS) values are the mean value of three replicates ± standard deviation. Values followed by different lowercase letters within a column are significantly different according to the least significant difference test (*p* < 0.05) using IBM SPSS Statistics 23.

## Data Availability

The sequences newly generated in this study have been submitted to the GenBank database.

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
