# Peer review of "Four New Species of Small-Spored Alternaria Isolated from Solanum tuberosum and S. lycopersicum in China"

_jof, 2023, doi:10.3390/jof9090880_

Round 1

Reviewer 1 Report

The study describes four new Alternaria species within section Alternaria, isolated from tomato and potato in China. A phylogenetic study and a morphological characterization were carried out in order to describe these new species. Also, a pathogenicity test was conducted to characterize the aggressiveness of the new strains to tomato and potato.

The authors highlight the importance of this finding to help disease management due to the aggressiveness of these new species to potato.

The description of these new species is of much importance for those studying Alternaria taxonomy and also for the characterization of potato and tomato pathogens and disease management.

However, there are some suggestions to improve the clarity of the information and the results obtained.

Specific comments:

1. Introduction section

Introduction section must be improved from line 50 to 57 since the authors make reference to the molecular identification of Alternaria species by phylogenetic analysis but they do not make reference to several papers that have shown that A. alternata and A. tenuissima cannot be distinguished as different species by phylogenetic analysis using the same gene regions used in this study or by phylogenomic analysis. Instead they claim that “molecular approaches have been utilized to identify Alternaria species, revealing the consistency between morphological classification and the clades determined by phylogenetic analysis” and they cite several papers that, from my point of view, are not actualized. I recommend reading these papers:

Andrew et al., 2009: An expanded multilocus phylogeny does not resolve morphological species within the small-spored Alternaria species complex. Mycologia 101, 95–109.

Woudenberg et al., 2015. Alternaria section Alternaria: Species, formae speciales or pathotypes? Stud. Mycol. 82, 1–21.

Armitage et al., 2015. Discrete lineages within Alternaria alternata species group: Identification using new highly variable loci and support from morphological characters. Fungal Biol. 119, 994–1006.

Armitage et al., 2020. Genomics, evolutionary history and diagnostics of the Alternaria alternata species group including apple and Asian pear pathotypes. Front. Microbiol. 10, 3124.

Luo et al., 2017. Identification of Alternaria species causing heart rot of pomegranates in California. Plant Dis. 101, 421–427.

Wang et al., 2021. Phylogenetic, morphological, and pathogenic characterization of Alternaria species associated with fruit rot of mandarin in California. Plant Dis. 105, 2606–2617.

Dettman and Eggertson, 2021. Phylogenomic analyses of Alternaria section Alternaria: A high-resolution, genome-wide study of lineage sorting and gene tree discordance. Mycologia, 113, 1218–1232.

Dettman and Eggertson, 2022. New molecular markers for distinguishing the main phylogenetic lineages within Alternaria section Alternaria. Can. J. Plant Pathol. https://doi.org/10.1080/07060661.2022.2061605

 2. Materials and methods section

The phylogenetic analysis section should be improved

For the Bayesian Inference, did you use the same substitution model? O did you determine it for each data set? Make the analysis description more clear.

3. Result section

Line 138: If most isolates (88.13%) belong to A. alternata species, why did you just used two strains for the phylogenetic analysis? Why from 118 small-spored Alternaria strains isolated you just analyzed 12 strains? You should clarify these in the text.

Line 139: You should clarify that the 88 strains in the phylogenetic analysis are comprised by reference strains and the studied strains because the way it is described it’s confusing.

Line 144: the authors say that the ML tree was similar to the RPB2 phylogram but they did not mention in the Materials and Methods section if they performed an individual phylogenetic analysis for each gene.

Lines 145 to 155: The results description from the phylogenetic analysis should be improved since the way the results are presented is very confusing. I recommend to use the word “close to” instead of “together with” to explain the distribution of the new species in the tree or the closeness to other Alternaria species. I think that the phylogenetic relationships are better describe in the “notes” from the new species description.

Figure 1: I think that the quality of the image should be improved and also the size since it’s very difficult to appreciate where the strains under study are and also the names from each strain used in the analysis. The support values are difficult to see.

Also, I think that the species under study in the clades from the tree should be without the species name, instead put Alternaria sp. And the strain name: Alternaria sp. YZU 221222 and then, next to each clade, as it is present in the figure, the new name or the species name. Because you are identifying them with this phylogenetic study. 

Lines 82 to 92: Please italicize all the scientific names and revise the rest of the manuscript.

4. Discussion section

Line 100 to 101: review considering the recommended reading mentioned above.

Line 147: The authors did not find any symptoms when the new species were inoculated in tomato leaves although some of them were isolated from symptomatic tomato samples. Can the authors give some explanation to this???

Line 151: The authors say that they did not find A. tenuissima in the study. However, several studies have reported that A. tenuissima and A. alternata cannot be separated as different species by phylogenetic studies and also that morphological characters’ overlap making their differentiation very difficult. They have proposed that A. tenuissima should be referred as A. alternata. So, I suggest that you include some A. tenuissima sequences from databases for the analyzed genes in the phylogenetic study in order to see how your A. alternata strains cluster with these sequences.

The quality of English should be improved since there are some sentences that are confusing. 

Author Response

Dear Editor:

It is my honor to have a chance to publish paper on Journal of fungi.

I would like to thank for all the comments. We revised manuscript ID: jof-2577294 entitled " Four new species of small-spored Alternaria isolated from Solanum tuberosum and S. lycopersicum in China" according to the comments and the revised areas were marked in the relevant places.

The responses are as follows:

Reviewer 1:

The study describes four new Alternaria species within section Alternaria, isolated from tomato and potato in China. A phylogenetic study and a morphological characterization were carried out in order to describe these new species. Also, a pathogenicity test was conducted to characterize the aggressiveness of the new strains to tomato and potato.

The authors highlight the importance of this finding to help disease management due to the aggressiveness of these new species to potato.

The description of these new species is of much importance for those studying Alternaria taxonomy and also for the characterization of potato and tomato pathogens and disease management.

However, there are some suggestions to improve the clarity of the information and the results obtained.

Specific comments:

  1. Introduction section

Introduction section must be improved from line 50 to 57 since the authors make reference to the molecular identification of Alternaria species by phylogenetic analysis but they do not make reference to several papers that have shown that A. alternata and A. tenuissima cannot be distinguished as different species by phylogenetic analysis using the same gene regions used in this study or by phylogenomic analysis. Instead they claim that “molecular approaches have been utilized to identify Alternaria species, revealing the consistency between morphological classification and the clades determined by phylogenetic analysis” and they cite several papers that, from my point of view, are not actualized. I recommend reading these papers:

Response: The recommended references were already ready earlier. Morphological species, Alternaria alternata and A. tenuissima are considered as one species A. alternaria based on genomic sequence and multiple gene loci (Woudenberg et al. 2015). About their classification, the both morph-species should be further study.

The mentioned sentence was revised. Please check the Line 47-52.

  1. Materials and methods section

1.The phylogenetic analysis section should be improved

For the Bayesian Inference, did you use the same substitution model? O did you determine it for each data set? Make the analysis description more clear.

Response: Yes, the best-fit model for the data was calculated using MrModelTest v. 2.3 and each data set was determined. It was improved. Please check the Line 101-109.

  1. Result section
  2. Line 138: If most isolates (88.13%) belong to A. alternata species, why did you just used two strains for the phylogenetic analysis? Why from 118 small-spored Alternaria strains isolated you just analyzed 12 strains? You should clarify these in the text.

Response: The main focus of this manuscript is the four new species. The two strains were selected based on pre-experiments and also used for a comparison with those four new taxa in pathogenicity. More work will be conducted among the A. alternata strains soon.

  1. Line 139: You should clarify that the 88 strains in the phylogenetic analysis are comprised by reference strains and the studied strains because the way it is described it’s confusing.

Response: It was revised. Please check the Line 132-133.

  1. Line 144: the authors say that the ML tree was similar to the RPB2 phylogram but they did not mention in the Materials and Methods section if they performed an individual phylogenetic analysis for each gene.

Response: It was revised. Please check the Line 96-97.

  1. Lines 145 to 155: The results description from the phylogenetic analysis should be improved since the way the results are presented is very confusing. I recommend to use the word “close to” instead of “together with” to explain the distribution of the new species in the tree or the closeness to other Alternaria species. I think that the phylogenetic relationships are better describe in the “notes” from the new species description.

Response: It was revised. Please check the Line 140-142.

  1. Figure 1: I think that the quality of the image should be improved and also the size since it’s very difficult to appreciate where the strains under study are and also the names from each strain used in the analysis. The support values are difficult to see.

Also, I think that the species under study in the clades from the tree should be without the species name, instead put Alternaria sp. And the strain name: Alternaria sp. YZU 221222 and then, next to each clade, as it is present in the figure, the new name or the species name. Because you are identifying them with this phylogenetic study. 

Response: The quality of the image was improved and the strain name was revised. Please check the Figure 1.

  1. Lines 82 to 92: Please italicize all the scientific names and revise the rest of the manuscript.

Response: The abbreviations for primer names are written in reference to woudenberg et al., where ITS and OPA10-2 primers should be in ortho, and other primers should be in italics.

(Woudenberg, J.H.C.; Seidl, M.F.; Groenewald, J.Z., De, V.M., Stielow, J.B., Thomma, B.P.H.J.; et al. Alternaria section Alternaria: species, formae speciales or pathotypes? Stud. Mycol. 2015, 82, 1–21. https://doi.org/10.1016/j.simyco.2015.07.001)

  1. Discussion section
  2. Line 100 to 101: review considering the recommended reading mentioned above.

Response: After further revision, we think the description of this sentence is accurate (Woudenberg et al. 2015)

  1. Line 147: The authors did not find any symptoms when the new species were inoculated in tomato leaves although some of them were isolated from symptomatic tomato samples. Can the authors give some explanation to this???

Response: It is possible that the spore found on the tomato is saprophytic or the tomato cultivar used for the test was the resistant variety. Please check the Line 317-320.

  1. Line 151: The authors say that they did not find A. tenuissima in the study. However, several studies have reported that A. tenuissima and A. alternata cannot be separated as different species by phylogenetic studies and also that morphological characters’ overlap making their differentiation very difficult. They have proposed that A. tenuissima should be referred as A. alternata. So, I suggest that you include some A. tenuissima sequences from databases for the analyzed genes in the phylogenetic study in order to see how your A. alternata strains cluster with these sequences.

Response: It is not proper to be described. In my opinion, A. tenuissima is better to be referred as A. alternata. Please check the Line 321-322.

Reviewer 2 Report

The research is relevant and interesting. The results contain a lot of new and valuable data, which makes it possible to publish the article in JoF. There are some small remarks that require corrections.

1. In the Sampling and isolation section and in the Table 1, it is necessary to write how many samples were collected and to clarify what one sample was in the context of this paper. How many fields do the samples represent?

2. The Latin names on page 15 should be italicized.

3. Grammar and spelling should be checked. I have suspicions that there are errors in some words and sentences. For example, I suggest checking the last sentence of the summary and also the word ‘typifification’.

Author Response

Dear Editor:

It is my honor to have a chance to publish paper on Journal of fungi.

I would like to thank for all the comments. We revised manuscript ID: jof-2577294 entitled " Four new species of small-spored Alternaria isolated from Solanum tuberosum and S. lycopersicum in China" according to the comments and the revised areas were marked in the relevant places.

The responses are as follows:

Reviewer 2:

The research is relevant and interesting. The results contain a lot of new and valuable data, which makes it possible to publish the article in JoF. There are some small remarks that require corrections.

  1. In the Sampling and isolation section and in the Table 1, it is necessary to write how many samples were collected and to clarify what one sample was in the context of this paper. How many fields do the samples represent?

Response: It was revised. Please check the Table 1.

  1. The Latin names on page 15 should be italicized.

Response: It was revised. Please check the Page 15.

  1. Grammar and spelling should be checked. I have suspicions that there are errors in some words and sentences. For example, I suggest checking the last sentence of the summary and also the word ‘typifification’.

Response: We have improved the last sentence of the summary. Please check the Line 16-19.

The word ‘typifification’ should be ‘typification’ and it was revised. Please check the Line 155, 185, 210, 236.